# Effectiveness of interventions for dementia in low- and middle-income countries: protocol for a systematic review, pairwise and network meta-analysis

Maximilian Salcher-Konrad,[1] Huseyin Naci,[2] David McDaid,[1] Suvarna Alladi,[3] Deborah Oliveira,[4,5] Andra Fry,[6] Shereen Hussein,[7] Martin Knapp,[1] Christine Wayua Musyimi,[8] David Musyimi Ndetei,[8,9] Mariana Lopez-Ortega,[10] Adelina Comas-Herrera[1]

For numbered affiliations see end of article.

**Correspondence to**
Maximilian Salcher-Konrad;
m.salcher@lse.ac.uk

## ABSTRACT

**Introduction** There are more people living with dementia in low- and middle-income countries (LMICs) than in high-income countries. Evidence-based interventions to improve the lives of people living with dementia and their carers are needed, but a systematic mapping of methodologically robust studies in LMICs and synthesis of the effectiveness of dementia interventions in these settings is missing.

**Methods and analysis** A systematic review and meta-analysis will be conducted to answer the question: Which dementia interventions were shown to be effective in LMICs and how do they compare to each other? Electronic database searches (MEDLINE, EMBASE, PsycINFO, CINAHL Plus, Global Health, WHO Global Index Medicus, Virtual Health Library, Cochrane CENTRAL, Social Care Online, BASE, MODEM Toolkit, Cochrane Database of Systematic Reviews) will be complemented by hand searching of reference lists and local knowledge of existing studies from an international network of researchers in dementia from LMICs. Studies will be eligible for inclusion if they were published between 2008 and 2018, conducted in LMICs and evaluated the effectiveness of a dementia intervention using a study design that supports causal inference of the treatment effect. We will include both randomised and non-randomised studies due to an anticipated low number of well-conducted randomised trials in LMICs and potentially greater external validity of non-randomised studies conducted in routine care settings. In addition to narrative synthesis of the interventions, feasibility of pairwise and network meta-analyses will be explored to obtain pooled effects of relative treatment effects.

**Ethics and dissemination** Secondary analysis of published studies, therefore no ethics approval required. Planned dissemination channels include a peer-reviewed publication as well as a website, DVD and evidence summaries.

**Prospero registration number** CRD42018106206.

### Strengths and limitations of this study

► This protocol defines the scope, inclusion/exclusion criteria and analytical approach for the first comprehensive assessment of methodologically robust dementia intervention studies in low- and middle-income countries (LMICs), the setting where most people with dementia currently live.

► Eligible study designs include both randomised trials and non-randomised studies supporting causal inference of treatment effects, with stringent eligibility criteria applied for the latter.

► Planned analyses include a narrative synthesis mapping out the interventions studied in LMICs as well as traditional pairwise and network meta-analysis, capable of yielding relative treatment effect estimates for interventions that have been compared either directly or indirectly (through a common comparator) to each other.

► A low number of studies may be eligible for inclusion, potentially limiting the scope for quantitative meta-analysis.

## INTRODUCTION

More people with dementia are now living in low- and middle-income countries (LMICs) compared with high-income countries.[1] Dementia is described by the Alzheimer's Association as 'an overall term that describes a group of symptoms associated with a decline in memory or other thinking skills severe enough to reduce a person's ability to perform everyday activities'.[2] Like other mental disorders, it places a significant burden on societies in LMICs, where care is often provided by family members and/or financed out-of-pocket, magnifying its impact beyond the individual living with

the condition.[3] While no curative treatment for dementia exists, a range of interventions aimed at improving the lives of people living with dementia and their carers have been developed and progress made in evaluating and understanding which of these are effective.[4] Despite the large and increasing burden of dementia in LMICs, these interventions have been primarily evaluated in high-income settings. Better understanding of the impact of dementia in LMICs and how people living with the disease can be better supported are therefore considered to be a priority.[5] Building capacity for research and policy-making is at the heart of the recently launched strengthening responses to dementia in developing countries (STRiDE) project (www.stride-dementia.org), as part of which a systematic review of the effectiveness of dementia interventions in LMICs will be conducted.

Research on dementia and dementia-related interventions, many aiming to improve the quality of life of those who are affected by the condition and their carers, has increased considerably since it was recognised as a key challenge for care systems: in mid-2018, the Cochrane Database of Systematic Review (www.cochranelibrary.com) listed over 120 systematic reviews of dementia-related interventions carried out since 2000 by their Dementia and Cognitive Improvement Group. Evidence on effectiveness and cost-effectiveness of a variety of dementia interventions has been recently summarised by other groups, prime among them the MODEM (modelling outcome and cost impacts of interventions for dementia) project.[6] A comprehensive database of over 1400 dementia intervention studies, along with a toolkit containing evidence summaries for decision-makers, people living with dementia and their carers was created (http://toolkit.modem-dementia.org.uk/).

The increase in knowledge about disease aetiology, prevention and management is expected to contribute to better quality of life for people living with dementia and their carers through the implementation of evidence-based approaches to diagnosing, managing and enhancing quality of life while living with the disease. The WHO,[5] organisations representing people living with dementia and their families,[7] and researchers[4] are all calling for decision-makers to focus on such approaches when developing policies and programmes. Evidence-based practices are informed by studies with strong research designs supporting a causal link between an intervention (be it a drug, non-pharmacological therapy, organisational change or another form of intervention) and improved outcomes in people living with dementia and their carers. While randomised controlled trials (RCTs) are considered the 'gold standard' of intervention studies, other study designs exist that allow researchers to draw causal conclusions about the effect of an intervention in the absence of RCTs and can provide essential evidence on 'what works'.[8 9]

Interventions with proven positive impacts in high-income countries have previously been summarised in the MODEM Dementia Evidence Toolkit. However, it is unclear whether interventions that showed beneficial effects in high-income settings, such as cognitive stimulation therapy for cognition and quality of life,[10 11] advance care planning,[12] training for formal carers (such as Staff Training in Assisted Living Residences (STAR))[13] and support for family carers (STRAtegies for RelaTives (START)),[14] are also effective in less-resourced settings, where awareness about dementia is lacking and dementia care may not be a policy priority. In this context, dementia is reported to be under-diagnosed and specialised care is often not available,[7] raising the question of which dementia interventions are effective in LMICs. Indeed, previously developed recommendations for packages of dementia care in LMICs were largely based on evidence from high-income countries,[7 15] indicating a need to better understand what works in LMICs. The aim of this systematic review is to help fill this research gap by identifying dementia interventions for which robust evidence on effectiveness in LMICs exists and to synthesise available evidence to determine which interventions have the most potential of achieving desired outcomes in these settings.

## METHODS AND ANALYSIS

This systematic review and meta-analysis protocol adheres to the Preferred Reporting Items for Systematic Review and Meta-Analysis Protocols checklist[16] and was registered on the PROSPERO platform (www.crd.york.ac.uk/prospero): CRD42018106206.

### Review question

Which dementia interventions have been shown to be effective in low- and middle-income countries and how do they compare to each other?

### Inclusion criteria

#### Population

We will include studies that have evaluated the effectiveness of an intervention in adults (18 years and over) living with dementia or their carers.

We will include studies in all stages (including early- to late-stage dementia) and different forms of dementia (eg, Alzheimer's disease, vascular dementia, Lewy bodies dementia and mixed dementias, as well as less common forms such as fronto-temporal dementia). We will also take into consideration risks of dementia associated with conditions such as HIV/AIDS that may be more prevalent in LMICs to ensure these are captured in our search strategy. Due to the particularly high risk of people with mild cognitive impairment (MCI) developing dementia (approximately one-third will develop dementia over a period of 3 to 10 years),[17] we will also include studies focusing on this group.

Carers of people living with dementia include family members, other unpaid carers, as well as professional carers, irrespective of whether or not they are paid, and

whether or not they are living with the person they care for.

Studies will be included if they were conducted in a country considered a LMIC according to the Organisation for Economic Co-operation and Development (OECD) categorisation at any time during the study period. The lists of aid recipient countries for the years 2008 to 2018 were used to identify LMICs (http://www.oecd.org/development/financing-sustainable-development/development-finance-standards/historyofdaclistsofaidrecipientcountries.htm).

We will include studies that were conducted entirely in LMICs, or (in the case of multi-country studies) where 50% or more of study participants received the intervention of interest in LMICs or where results were presented separately for participants in LMICs.

### Intervention

One of the aims of this systematic review is to identify which interventions have been rigorously evaluated in LMICs. The eligibility criteria for types of interventions are therefore deliberately kept open: we will include all interventions that aim to improve the lives of people living with dementia and their carers contingent on the intervention having been subjected to a rigorous evaluation (see eligible study designs below). We will apply an 'effectiveness' perspective with respect to interventions, that is, interventions are eligible when defined as such in a study and improved outcomes are expected through the intervention.[18] We will focus on people already living with dementia or MCI and their carers and will therefore exclude primary prevention studies (ie, prevention of dementia in people without cognitive impairment).

Dementia interventions can vary in terms of who they are targeting (person living with dementia, unpaid carers, care professionals and care systems), what the aim of the intervention is (secondary prevention, treatment, disease management, coping with caring for people living with dementia, managing the impact of dementia on the care system) and their mechanism (including diagnosis; pharmacological treatment; cognitive therapy; technological interventions; training; exercise; sleep therapy; music therapy; organisation of care, including advance care directives and case management; support for carers; financing of care; policy interventions and others).[4]

### Comparison

All comparisons will be eligible, including active comparators, usual practice/standard care/placebo and no action.

### Outcomes

Outcomes of dementia intervention studies are highly heterogeneous and may vary by who is affected (person living with dementia, their carer(s), wider society and care system) and type of outcome (clinical outcomes, such as cognitive, neurological, psychological, psychosocial; quality of life and functioning outcomes; care and delivery of care, such as the use of feeding tubes, hospitalisations, institutionalisations; economic outcomes; diagnosis rates; knowledge of the disease and ability to cope with caring for people with dementia). For example, recommendations on dementia interventions were recently made based on a review of the evidence on outcomes in domains ranging from, among others, cognition and neuropsychiatric outcomes to end-of-life care and care delivery.[4]

Given the broad range of potentially relevant outcomes, a search strategy that captures all of these is not feasible. Restricting inclusion criteria to certain types of outcomes would risk excluding potentially important parts of the evidence base for dementia interventions in LMICs. The impact of dementia is recognised to be experienced by different people and at different levels, including by people living with dementia, their carers, communities and the care system and wider society,[19] and interventions therefore need to take all of these perspectives into account.[5]

### Study design

The research question we aim to answer is what works in dementia in LMICs. We are therefore interested in the effect that an intervention had on outcomes in people receiving the intervention, their carers and the wider care system.[20] Accordingly, we will apply a causal inference framework in this review: study designs eligible for inclusion are those that support a causal link between the intervention and observed outcomes. This includes experimental designs (RCTs and cluster-RCTs) as well as non-randomised designs suitable for causal inference, defined as 'comparisons of 'potential outcomes' (…) under different treatment conditions on a common set of units'.[8] We will include both randomised and non-randomised studies due to an anticipated low number of well-conducted randomised trials in LMICs, based on previous work (searches conducted for the MODEM Evidence Toolkit, and a previous review on packages of care for people living with dementia in LMICs[15]). Also, non-randomised studies may have greater generalisability beyond the studied environment when conducted in settings and populations closer resembling routine practice.

In order to maximise the number of eligible studies and synthesise as much robust evidence about dementia interventions in LMICs as possible, we will extend the eligibility criteria used by the Cochrane Effective Practice and Organisation of Care (EPOC) Group[21] to also include other quasi-experimental study designs in addition to those typically seen in methodologically robust evaluation studies (non-randomised controlled trials, controlled before-after studies and interrupted time series studies). The label 'quasi-experimental' is discipline-specific and members of the Cochrane Non-Randomised Studies for Interventions Methods Group therefore developed a 'label-free' taxonomy of quasi-experimental studies that aims to define such studies through a series of questions

**Table 1**  Checklist for eligible study designs (based on Reeves *et al*[22])

| Feature | Eligible for inclusion if… |
|---|---|
| **1. Was the intervention/comparator:** | |
| Allocated to (provided for/administered to/chosen by) individuals? | Yes |
| Allocated to (provided for/administered to/chosen by) clusters of individuals? | Yes |
| Clustered in the way it was provided (by practitioner or organisational unit)? | Yes |
| **2. Were outcome data available:** | |
| After intervention/comparator only (same individuals)? | No |
| After intervention/comparator only (not all same individuals)? | No |
| Before (once) AND after intervention/comparator (same individuals)? | Yes |
| Before (once) AND after intervention/comparator (not all same individuals)? | Yes |
| Multiple times before AND multiple times after intervention/comparator (same individuals)? | Yes |
| Multiple times before AND multiple times after intervention/comparator (not all same individuals)? | Yes |
| **3. Was the intervention effect estimated by:** | |
| Change over time (same individuals at different time points)? | Yes |
| Change over time (not all same individuals at different time points)? | Yes |
| Difference between groups (of individuals or clusters receiving either intervention or comparator)? | Yes |
| **4. Did the researchers aim to control for confounding (design or analysis)** | |
| Using methods that control in principle for any confounding? | Yes |
| Using methods that control in principle for time-invariant unobserved confounding? | Yes |
| Using methods that control only for confounding by observed covariates? | No |
| **5. Were groups of individuals or clusters formed by** | |
| Randomisation? | Yes |
| Quasi-randomisation? | Yes |
| Explicit rule for allocation based on a threshold for a variable measured on a continuous or ordinal scale or boundary (in conjunction with identifying the variable dimension, below)? | Yes |
| Some other action of researchers? | Yes |
| Time differences? | Yes |
| Location differences? | Yes |
| Healthcare decision makers/practitioners? | Yes |
| Participants' preferences? | Yes |
| Policymaker? | Yes |
| On the basis of outcome? | No |
| Some other process? (specify) | - |
| **6. Were the following features of the study carried out after the study was designed** | |
| Characterisation of individuals/clusters before intervention? | Yes |
| Actions/choices leading to an individual/cluster becoming a member of a group? | Yes |
| Assessment of outcomes? | Yes |
| **7. Were the following variables measured before intervention:** | |
| Potential confounders? | Yes |
| Outcome variable(s)? | Yes |

in six domains.[22] We will use their checklist to guide the selection of studies by study design (see table 1). Since we anticipate a shortage of methodologically robust studies of dementia interventions in LMICs, but still aim to synthesise the best available evidence that supports causal inference of treatment effects, we adapted the original checklist by removing restrictions to answer 'yes' to only one question in domains 2, 3 and 4. This will enable us to include several strong experimental and quasi-experimental study designs, including RCTs (including cluster-RCTs), non-randomised controlled trials, controlled before-after studies, interrupted time series studies,

difference-in-differences studies, instrumental variables studies, regression discontinuity design studies and other study designs that employ methods such as propensity score matching that attempt to control for observed and unobserved confounders.

As a contingency plan in case no such studies can be identified in LMICs, we define a secondary set of inclusion criteria to still capture intervention studies in LMICs with less robust designs but that could still inform our understanding of what works in dementia in LMICs in the absence of better study designs. The secondary set of inclusion criteria will mirror the primary inclusion criteria, with the exception of study design. In the secondary set of inclusion criteria, studies will also be eligible for inclusion if they control only for observed confounders (criterion 4 in table 1).

### Publication type

We will include peer-reviewed journal articles, including primary publications of intervention studies and systematic reviews of these, and grey literature describing evaluations of dementia interventions in LMICs (eg, PhD theses and reports published by care system administrative bodies and non-governmental organisations). Systematic reviews are only eligible for inclusion if their focus is on LMICs (ie, if LMICs were specified as geographical setting in their inclusion criteria) and we will use these to identify primary studies eligible for inclusion (eligible if 50% or more participants were in LMICs or results were reported separately for LMIC settings). We will also include conference abstracts, provided they contain information to assess eligibility for inclusion.

### Exclusion criteria

Exclusion criteria are listed in box 1.

### Search strategy

We will conduct searches in bibliographic databases (box 2), using text words, subject headings and other search functions that each database offers. We will limit the searches to studies published from 2008 to 2018 and, where possible, will use filters to exclude editorials, commentaries and letters to the editor, as well as running a search filter for animal studies.

We will also make use of previous efforts to systematically capture studies evaluating the effectiveness of dementia interventions and will review studies identified by other authors for inclusion. First, we will review studies included in the MODEM Dementia Evidence Toolkit (http://toolkit.modem-dementia.org.uk). The MODEM Toolkit contains over 1400 primary studies of dementia interventions. While the primary focus of the toolkit was on high-income settings, studies from LMICs may be included in the database. Second, we will screen existing Cochrane systematic reviews of dementia interventions for primary studies conducted in LMICs. We will review all Cochrane systematic reviews indexed in the Cochrane Library's dementia and cognition topic.

---

**Box 1    Exclusion criteria**

We will exclude studies meeting any of the following conditions:
► Studies where less than 50% of participants received the intervention of interest in low- and middle-income countries (LMICs) or where no results are available for a LMIC subgroup.
► Studies where no clear intervention was described.
► Studies of associations between exposure and outcome (as opposed to causal links between intervention and outcome).
► Studies of primary prevention of dementia.
► Studies where reports are not available in a language spoken by the international project team members (languages spoken include Arabic, Chinese, English, French, German, Bahasa Indonesia, Portuguese, Romanian, Spanish, Turkish and others).
► Studies that do not attempt to control for unobserved confounding, including cohort studies, case-control studies, cross-sectional studies and case series (unless these have applied methods to control for unobserved confounders).
► Narrative reviews, overview articles, editorials, commentaries, letters to the editor.
► Studies where animals (as opposed to humans) received the intervention.

In case no studies meeting our primary inclusion criteria for study design can be identified, we will apply secondary inclusion criteria as specified under 'Study design' and will amend the exclusion criteria for study design accordingly.

---

In addition to database searches, we will scan the reference lists of included studies to capture potentially missed ones. We will use Google Scholar, Scopus and Web of Science to carry out citation searches for the included papers.

We will also capitalise on the expertise of STRiDE consortium partners (researchers and dementia advocacy groups) in seven LMICs (Brazil, India, Indonesia,

---

**Box 2    Online databases to be searched**

► MEDLINE (via OVID).
► EMBASE (via OVID).
► PsycINFO (via OVID).
► CINAHL Plus (via EBSCO).
► Global Health (via CABI).
► WHO Global Index Medicus (GIM, includes databases from the six WHO regions: AIM (AFRO), LILACS (AMRO/PAHO), IMEMR (EMRO), IMSEAR (SEARO), WPRIM (WPRO)).
► Cochrane CENTRAL and Cochrane Database of Systematic Reviews.
► Social Care Online.
► BASE.
► Virtual Health Library.

In addition to the international databases listed above, we will consider additional national or regional databases through consultation with strengthening responses to dementia in developing countries (STRiDE) partners in seven low- and middle-income countries (Brazil, India, Indonesia, Jamaica, Kenya, Mexico, South Africa). These databases might not be available in English and we will capitalise on the variety of languages spoken in the STRiDE consortium to include these in our search strategy, if deemed of sufficient added value for the identification of studies meeting our inclusion criteria.

---

Jamaica, Kenya, Mexico, South Africa), and an expanded network of STRiDE collaborators in other countries, to include published and unpublished studies from these and other countries meeting our eligibility criteria if they were not captured by database searches.

We will further search trials registers (ClinicalTrials.gov and the International Clinical Trials Registry Platform) to identify ongoing or planned clinical trials of dementia interventions in LMICs.

We will capitalise on the expertise of information specialist support team of London School of Economics and Political Science (LSE) to finalise the search strategy.

### Search terms

Search terms were selected to reflect the key concepts making up the review question: dementia interventions that have been studied using robust research designs in LMICs. Exemplary search terms for MEDLINE via OVID for the concepts of population and intervention, study design and LMICs are provided in table 2, including the number of hits for each search as per 10 October 2018. The search for MEDLINE will be translated for the other databases.

The 'intervention' concept provided in table 2 (lines 22 to 98) includes both generic terms as well as a range of specific interventions that have been studied for dementia before, although usually in high-income countries. We based the list of specific interventions on a thorough review of the evidence, conducted as part of the Lancet commission,[4] and through revision by the international STRiDE consortium. While this might not be a comprehensive list of all interventions that have ever been studied for dementia, we aimed to increase the likelihood of identifying relevant studies by combining generic search terms pertaining to interventions, therapies, etc with specific intervention types and names of interventions, such as drug names.

We will use established study design and geographical location search filters to narrow down the number of results and will make adaptations as necessary to reflect our inclusion criteria. If such filters are not available for some databases, we will consider adapting the search filters used for MEDLINE.

The search filters for eligible study designs are provided in table 2, rows 101 to 108 (for RCTs), 110 to 121 (for cluster-RCTs) and 123 to 147 (for other quasi-experimental studies), respectively. In order to identify RCTs, we will use the sensitivity-maximising version of the highly sensitive filter for RCTs recommended in the Cochrane Handbook.[23] For identification of cluster-RCTs, we will use the sensitivity-maximising filter for cluster-RCTs developed by Taljaard et al.[24] This filter was validated, among others, for a sample of cluster-RCTs in sub-Saharan Africa from before 2001 and showed good sensitivity (94%), with overall higher sensitivity for more recent studies. For identification of non-randomised studies that support causal inference of the relationship between exposure and outcome, we created an additional set of search terms pertaining

to commonly used labels for quasi-experimental studies. We enhanced these search terms with the 'aetiology' filter developed by the Hedges Group.[25] The methodological criteria used for developing the filter were found to be a good match for our inclusion criteria (including the use of a clearly identified comparison group, with examples given by the authors including 'randomised controlled trials (RCT), quasi-randomised controlled trials, non-randomised controlled trials, cohort studies with case-by-case matching or statistical adjustment to create comparable groups or nested case–control studies'[25]).

Finally, rows 149 to 157 of table 2 show the exemplary search terms for LMICs, based on a filter for LMICs that is available from the Cochrane EPOC website (http://epoc.cochrane.org/lmic-filters). The search terms were expanded and updated to reflect the list of LMICs from 2008 to 2018.

Filters for LMICs and study designs were tested in MEDLINE for a sample of 10 studies (six selected for being conducted in LMICs,[26–31] four selected for using quasi-experimental study designs[32–35]). The sensitivity of the search strategy for this sample was 100% (10/10 studies) for searches with study design filters, and 100% (6/6 studies) when adding the LMIC filter.

### Study selection

Search results from the database searches will be exported to a bibliographic reference manager and deduplicated. Two researchers will independently screen articles at the title and abstract level for eligibility. Full text for articles deemed eligible at this stage will be retrieved, including articles where no abstract was available at the first screening. Articles describing the same study will be linked together. Full-text articles will be independently assessed for inclusion eligibility by two researchers. Deviating decisions on inclusion will be resolved by discussion and consensus between the two researchers. If no consensus is reached by the two researchers, we will consult with a third researcher for a final decision on inclusion. We will record rates of agreement between researchers independently screening abstracts and full texts to measure the extent of disagreement using the kappa statistic. We will take a kappa of >0.8 as indication of 'very good' agreement.[36]

Inclusion and exclusion of studies at each stage will be illustrated with a Preferred Reporting Items for Systematic Reviews and Meta-Analyses flow chart.[37] A list of included and excluded studies (with primary exclusion reason) will be made available.

### Data extraction

Two researchers will independently extract the information listed in box 3 from included studies. Selection of data to be extracted was informed by the review question and by practical insights regarding data extraction and synthesis from non-randomised studies.[38 39] We will contact authors of included abstracts to obtain full details

**Table 2** Exemplary search terms (MEDLINE via OVID)

| Search # | Terms used | No. of hits |
|---|---|---|
| 1 | Exp Dementia/ | 148649 |
| 2 | Dement*.mp | 114488 |
| 3 | Amentia*.mp | 102 |
| 4 | (major adj3 cognit* adj3 disorder*).mp | 159 |
| 5 | (alzheimer* or alzeimer* or (cortical adj4 sclerosis)).mp | 140775 |
| 6 | ((encephalopath* or cognit* or neurocogniti*) adj4 (aids or acquired immun?deficiency syndrome or acquired immun* or neurocogniti*) adj4 (aids or acquired immun?deficiency syndrome or hiv or human immun?deficiency virus or human immun? deficiency virus)).mp | 3302 |
| 7 | 'Pick Disease of the Brain'/ | 495 |
| 8 | (Pick* disease).mp | 3308 |
| 9 | (lobar adj3 atroph* adj3 brain).mp | 10 |
| 10 | Huntington Disease/ | 11069 |
| 11 | (Huntington* disease).mp | 16193 |
| 12 | (Huntington* chorea).mp | 1349 |
| 13 | Lewy Body Disease/ | 2761 |
| 14 | (Lewy bod* adj3 disease).mp | 4240 |
| 15 | (cerebr* adj3 deteriorat*).mp | 361 |
| 16 | (cerebr* adj3 insufficien*).mp | 2079 |
| 17 | ((frontotemporal or fronto temporal or corticobasal or cortico basal or frontal lobe*) adj4 (degenerati* or dysfunction*)).mp | 5217 |
| 18 | ((cognit* or memory or cerebr*) adj3 (declin* or impair* or los* or deteriorat* or degenerat* or insufficen*)).mp | 106843 |
| 19 | Cognitive Dysfunction/ | 8749 |
| 20 | (MCI or (mild adj2 cognit* impair*)).ti;ab. | 20332 |
| **21** | **Or/1–20** | 314147 |
| **22** | **(Intervention* or therap* or treatment* or program* or manage* or prevent* or diagnos* or polic*).mp** | **12 485 776** |
| 23 | exp Cognitive Therapy/ | 23692 |
| 24 | (cognit* adj3 therap*).mp | 30470 |
| 25 | (cognit* psycho therap* or cognit* psychotherap*).mp. | 119 |
| 26 | (cognit* adj3 training).mp | 3571 |
| 27 | (cognit* adj3 rehab*).mp | 1911 |
| **28** | **Or/23–27** | 36923 |
| 29 | exp Drug Therapy/ | 1 246 404 |
| 30 | (Drug* or medicine* or pharmacotherap* or pharmaco* therap*).ti;ab. | 1 838 638 |
| 31 | exp Cholinesterase Inhibitors/ | 47229 |
| 32 | cholinesterase agent*.mp | 30 |
| 33 | cholinesterase inhibitor*.mp | 21340 |
| 34 | exp Antipsychotic Agents/ | 115322 |
| 35 | (Tranquil* adj3 (agent* or drug*)).mp | 12096 |

Continued

**Table 2** Continued

| Search # | Terms used | No. of hits |
|---|---|---|
| 36 | Antipsychotic*.mp | 63916 |
| 37 | (neuroleptic adj3 (agent* or drug*)).mp | 3296 |
| 38 | exp Serotonin Uptake Inhibitors/ | 34982 |
| 39 | (serotonin uptake inhibitor* or serotonin reuptake inhibitor*),mp or ssri*.ti,ab | 25984 |
| 40 | exp Benzodiazepines/ | 62786 |
| 41 | Benzodiazepine*.mp | 42836 |
| 42 | exp 'Hypnotics and Sedatives' / | 115358 |
| 43 | (Sedative adj3 (effect* or agent*)).mp | 5467 |
| 44 | (memantine or donepezil or rivastigmine or galantamine or souvenaid or risperidone or haloperidol or olanzapine or quetiapine or citalopram or dextromethorphan or carbamazepine or mirtazapine or sertraline or moclobemide or trazodone or melatonin or ramelteon or methylphenidate).mp | 105057 |
| *45* | *Or/29–44* | *3 080 487* |
| 46 | exp Exercise Therapy/ | 42856 |
| 47 | Exercis*.mp | 335419 |
| 48 | (Physical activit* or physical training).mp. | 99409 |
| 49 | (Aerobic* or arobic*).mp | 81338 |
| 50 | exp Exercise/ | 166405 |
| *51* | *Or/46–50* | *497 630* |
| 52 | (social adj3 activit*).mp | 9247 |
| 53 | (social adj3 engag*).mp | 2774 |
| 54 | (social adj3 stimul*).mp | 2178 |
| *55* | *Or/52–54* | *14 003* |
| 56 | exp Psychotherapy/ | 180508 |
| 57 | (psycholog* therap* or psychotherap*).mp | 85389 |
| 58 | ((behavio?r* adj3 therap*) or (conditioning adj3 therap*)).mp | 47185 |
| 59 | exp Counseling/ | 40343 |
| 60 | Counsel?ing.mp | 89908 |
| 61 | psychosocial support systems/ | 145 |
| 62 | ((Psychosocial or psycho social) adj3 (support or interven* or care)).ti,ab | 11193 |
| *63* | *Or/56–62* | *315 606* |
| 64 | exp Complementary Therapies/ | 208840 |
| 65 | ((Alternative or compl?ment* or traditional) adj3 (medicine* or therap*)).mp | 101303 |
| 66 | Acupunct*.mp | 25265 |
| 67 | (herb* adj3 (tea or remedy or remedies or medicine*)).ti,ab. | 16075 |
| 68 | Gingko.ti,ab | 346 |
| 69 | homeopath*.mp | 6178 |
| 70 | ((music or art or aroma or light or photo or pet or pets) adj3 therap*).ti,ab | 16656 |

Continued

**Table 2** Continued

| Search # | Terms used | No. of hits |
|---|---|---|
| 71 | Massage.ti,ab | 9025 |
| 72 | (mind adj3 body).ti,ab | 4035 |
| 73 | Phototherapy/ | 7273 |
| **74** | **Or/64–73** | 302 897 |
| 75 | exp Advance care planning/ | 8277 |
| 76 | (Advance? adj3 (care or medical or healthcare) adj3 plan*).mp | 3552 |
| 77 | (decision* adj3 (aid* or support)).mp | 41 646 |
| **78** | **Or/75–77** | 46 897 |
| 79 | Case Management/ | 9516 |
| 80 | (communicati* adj3 skill* adj3 training).mp | 1283 |
| 81 | (dementia care adj3 map*).mp | 92 |
| 82 | ((person* or patient*) adj3 cent* adj3 care).mp | 23 552 |
| **83** | **Or/79–82** | 34 244 |
| 84 | ('Resources for Enhancing Alzheimer's Caregiver Health' or 'Strategies for Relatives').ti,ab. | 88 |
| 85 | Caregivers/ed (Education) | 2484 |
| 86 | CAREGIVERS/px (Psychology) | 18 509 |
| 87 | Self-Help Groups/ | 8626 |
| 88 | exp Social support/ | 63 885 |
| **89** | **Or/84–88** | 87 559 |
| 90 | Diagnosis, Computer-Assisted/ | 21 241 |
| 91 | Telemedicine/ | 17 577 |
| 92 | (Telemedicine or tele medicine).mp | 21 573 |
| 93 | exp Computers, Handheld/ | 5107 |
| 94 | ((smart adj2 (phone* or device* or tablet*)) or smartphone*).mp | 8280 |
| 95 | cognitive aid*.mp. | 138 |
| 96 | Reminder*.ti,ab | 10 667 |
| 97 | Robot*.mp | 38 250 |
| **98** | **Or/90–97** | 102 392 |
| **99** | **22 or 28 or 45 or 51 or 55 or 63 or 74 or 78 or 83 or 89 or 98** | 13 800 589 |
| **100** | **21 and 99** | 208 431 |
| 101 | randomized controlled trial.pt | 462 693 |
| 102 | controlled clinical trial.pt | 92 457 |
| 103 | (randomized or randomised).ab | 495 756 |
| 104 | Placebo.ab | 189 758 |
| 105 | drug therapy.fs | 2 024 910 |
| 106 | Randomly.ab | 292 598 |

Continued

**Table 2** Continued

| Search # | Terms used | No. of hits |
|---|---|---|
| 107 | Trial.ab | 431 012 |
| 108 | Groups.ab | 1 806 682 |
| 109 | Or/101–108 | 4 294 404 |
| 110 | randomized controlled trial.pt | 462 693 |
| 111 | (Cluster* adj2 randomi*).tw | 9224 |
| 112 | ((communit* adj2 intervention*) OR (communit* adj2 randomi*)).tw | 6858 |
| 113 | Group* randomi*.tw | 2846 |
| 114 | Or/111–113 | 18 443 |
| 115 | intervention?.tw | 782 326 |
| 116 | cluster analysis/ | 56 328 |
| 117 | health promotion/ | 66 499 |
| 118 | program evaluation/ | 56 711 |
| 119 | health education/ | 57 721 |
| 120 | Or/115–119 | 965 052 |
| 121 | 114 or 120 | 968 364 |
| 122 | 110 or 121 | 1 366 397 |
| 123 | risk.mp. | 2 178 921 |
| 124 | exp cohort studies/ | 1 751 258 |
| 125 | between group.tw | 21 162 |
| 126 | Or/123–125 | 3 451 482 |
| 127 | (non random* or nonrandom*).mp | 33 945 |
| 128 | Non-Randomized Controlled Trials as Topic/ | 359 |
| 129 | quasi.mp | 41 169 |
| 130 | (natural adj3 experiment).mp | 1724 |
| 131 | instrumental variable*.mp | 1969 |
| 132 | Cohort.mp | 516 750 |
| 133 | before-after.mp | 4519 |
| 134 | (before adj2 after adj study).mp. | 1810 |
| 135 | Controlled Before-After Studies/ | 329 |
| 136 | (difference-in-difference* or diff-in-diff).mp | 1782 |
| 137 | regression discontinuity.mp | 223 |
| 138 | Historically Controlled Study/ | 139 |
| 139 | historical* control.mp | 3400 |
| 140 | Interrupted Time Series Analysis/ | 435 |
| 141 | Interrupted Time Series.mp | 2280 |
| 142 | Case-control.mp | 287 943 |

Continued

**Table 2** Continued

| Search # | Terms used | No. of hits |
|---|---|---|
| 143 | Case-Control Studies/ | 249334 |
| 144 | Match*.mp | 419961 |
| 145 | Propensity.mp | 47848 |
| 146 | Propensity Score/ | 5596 |
| 147 | Or/127–146 | 1 205 777 |
| *148* | *126 or 147* | *4 094 244* |
| 149 | Developing Countries.sh,kf. | 81106 |
| 150 | (Africa or Asia or Caribbean or West Indies or South America or Latin America or Central America).hw,kf,ti,ab,cp. | 242209 |
| 151 | (Afghan* or Albania* or Algeria* or Angola* or Anguilla* or Antigua* or Barbuda* or Argentin* or Armenia* or Azerbaijan* or Azeri or Bangladesh* or Barbad* or Benin* or Byelarus or Byelorussian or Belarus* or Belorussia* or Belize* or Bhutan* or Bolivia* or Bosnia* or Herzegovin* or Hercegovin* or Botswana or Motswana or Batswana or Brasil* or Brazil* or Burkina Faso or Burkina Fasso or Burkina* or Burundi* or Urundi* or Cambodia* or Cameroon* or Cameron* or Cabo Verde or Central African Republic or Chad* or Tchad* or Chile* or China or Chinese or Colombia* or Columbia* or Comoros or Comoro Islands or Comores or Comoran or Mayotte or Congo* or Costa Rica* or Cote d'Ivoire or Ivory Coast or Ivorian* or Cook Islands or Cuba* or Croat* or Djibouti* or Dominica* or East Timor or East Timur or Timor Leste or Timorese or Ecuador* or Equador* or Egypt* or El Salvador or Salvadoran or Eritrea* or Ethiopia* or Fiji* or Gabon* or Gambia* or Gaza or Georgia Republic or Georgian or Abkhazia* or Abchasia* or South Ossetia* or Ghana* or Grenada or Grenadian or Guatemala* or Guinea* or Guinea Bissau or Guian* or Guyana or Haiti* or Hondura* or India or Indian or Indonesia* or Iran* or Iraq* or Jamaica* or Jordan* or Kazakhstan* or Kazakh or Kenya* or Kiribati or Korea* or Kosovo or Kosova* or Kyrgyzstan or Kirghizia or Kyrgyz or Kirghiz or Kirgizstan or Lao PDR or Laos or Laotian or Lebanon or Lebanese or Lesotho or Mosotho or Basotho or Liberia* or Libya* or Macedonia* or FYROM or Madagasca* or Malagasy or Malaysia* or Malaya* or Malay or Sabah or Sarawak or Malawi* or Maldives or Maldivan or Mali or Malian or Marshall Islands or Marshallese or Mauritania* or Mauriti* or Agalega Islands or Mexico or Mexican or Micronesia* or Middle East* or Moldova* or Moldovia* or Transnistria* or Mongolia* or Montenegr* or Montserrat* or Morocc* or Mozambique or Mozambican or Myanmar* or Myanma or Burma or Burmese or Namibia* or Nauru* or Niue or Nepal* or Nicaragua* or Niger or Nigerien or Nigeria* or Pakistan* or Palau* or Palestine or Palestinian or Panama or Panamanian or Paraguay* or Papua New Guinea* or Peru or Peruvian or Philippines or Phillipines or Philipines or Filipino or Philipino or Phillipino or Rwanda* or Ruanda* or Saint Helen* or St Helen* or Samoa* or Seychell* or Seychell* or Sierra Leone* or Saint Kitts or St Kitts or Kittian or Nevis* or Saint Lucia* or St Lucia* or Ceylon or Sri Lanka* or Saint Vincent or Vicentian or Grenadines or Serbia* or Serb* or South Africa* or South African* or Sudan or Sudanese or Surinam* or Swaziland or Swazi or Eswatini or Syrian or Syria or Tajik or Tadzhik or Tadzhikistan or Tadjikistan or Tajikistan or Tadjik or Tanzania* or Thailand or Thai or Togo or Togolese or Tonga* or Tunisia* or Tokelau or Trinidad* or Tobago* or Turkey or Turkish or Turks or Turkmenistan or Turkmen or Tuvalu* or Uganda* or Ukraine or Ukrainian or Uruguay* or Uzbekistan* or Uzbek or Vanuatu or Venezuela* or Vietnam* or Viet Nam or Wallis Futuna or Yemen* or Zambia* or Zimbabwe*).hw,kf,ti,ab,cp. | 2 805 600 |
| 152 | ((developing or less* developed or under developed or underdeveloped or middle income or low* income or underserved or under served or deprived or poor*) adj (countr* or nation? or population? or world)).ti,ab. | 85468 |
| 153 | ((developing or less* developed or under developed or underdeveloped or middle income or low* income) adj (economy or economies)).ti,ab. | 440 |
| 154 | (low* adj (gdp or gnp or gross domestic or gross national)).ti,ab. | 217 |
| 155 | (low adj3 middle adj3 countr*).ti,ab. | 11181 |
| 156 | (lmic or lmics or third world or lami countr*).ti,ab. | 5792 |
| 157 | transitional countr*.ti,ab. | 146 |
| *158* | *Or/149–157* | *2 945 907* |
| 159 | 100 and (109 or 122 or 148) and 158 | 8278 |
| 160 | limit 159 to (comment or editorial or letter) | 74 |
| 161 | 159 not 160 | 8204 |
| 162 | limit 161 to yr="2008 -Current" | 6322 |
| 163 | exp animals/not humans.sh. | 4 496 553 |
| *164* | *162 not 163* | *5786* |

## Box 3 Data to be extracted from included studies

- ► Publication details.
- ► Source of funding for the study.
- ► Geographical location.
- ► Care setting.
- ► Study design: description and coding of the study design and how causality of a treatment effect was supported, including which co-variates were used in the analysis to control for confounding and judgement about whether criteria for causality for the specific study design were met (in the case of non-randomised studies).
- ► Participant details, including type of dementia and representativeness of local/regional/national population with dementia.
- ► Intervention: brief description of the intervention in terms of its aims, implementation and intervention details, including for example, duration and intensity of the intervention, dosage of drugs, existence of a protocol or manual for psychosocial, training or education interventions and other details that allow an informed judgement about the comparability of interventions within the same type of treatment.
- ► Comparator: description of the comparison group and the intervention received by them (if any), including a note on statistically significant differences in baseline characteristics between experimental and control groups.
- ► Outcomes: primary and secondary outcomes of the study, including information on how they were measured (instruments used).
- ► Results: effect size and measure of its variance for the primary outcome and for any other outcomes mentioned in the abstract or executive summary of the study. Preference will be given to adjusted effect sizes (ie, taking into account covariates that might not be balanced across experimental and control group), and in cases where several adjusted results are presented we will extract the one where selection bias is best controlled (either through design or analysis, eg, inclusion of most relevant confounding variables).
- ► Risk of bias information.

of their studies if the information is not available from abstracts.

### Risk of bias

We will assess the internal validity of included studies using appropriate tools. For RCTs, we will use the Cochrane Collaboration's recently updated risk of bias tool (RoB 2.0).[40]

We will use the Cochrane Collaboration's ROBINS-I tool to assess risk of bias in non-randomised studies.[41]

### Data synthesis

We will first describe the interventions that have been evaluated in rigorous study designs in LMICs and summarise the features of the intervention, where they were studied, characteristics of the studies and their findings. We are planning to tabulate interventions according to who they are targeted at (person living with dementia, their carers, care professionals and care systems), aim of the intervention (secondary prevention, treatment, disease management, coping with the disease and caring for people living with dementia, managing the impact of dementia on the care system), intervention type (including, but not limited to, pharmaceuticals, cognitive therapy, technological interventions, exercise, training,

diagnosis, organisation of care, financing of care, policy interventions) and outcomes studied. We plan to review details extracted from included studies on the population, intervention and outcomes (see box 3) to group studies accordingly. For example, in terms of population, included studies may be grouped according to severity of impairment (for people living with dementia) and type of carer (unpaid, paid, professional, family caregiver). Tabulation along this dimension will allow us to explore which types of interventions were shown to be effective for each of the groups in order to provide sufficiently granular findings to inform decision-making in specific contexts. Similarly, we plan to tabulate studies according to the aim of the intervention, intervention type and outcome studied, thereby providing a summary of the evidence for different aspects of dementia interventions.

Potentially relevant outcomes for this systematic review can be characterised by their type and the stakeholder group they refer to. A draft list of potentially relevant outcomes and their stakeholder groups according to which our analysis can be structured is provided in the online supplementary table. The list is subject to change after reviewing the outcomes used in included studies, which might use other outcomes that could necessitate a different categorisation.

We will use the GRADE approach to rate the quality of evidence for each intervention.[42]

### Quantitative synthesis

We will explore the feasibility of conducting quantitative synthesis of treatment effects through traditional pairwise and network meta-analysis. Feasibility of quantitative synthesis will be assessed for each intervention studied in our sample of included studies. We will assess the similarity of the specific intervention in each study with other candidate interventions for a meta-analysis, the participants in the studies where this intervention was evaluated and whether the same outcome was used. Quantitative synthesis can only be conducted for studies reporting the same outcome, although individual studies may use different instruments to measure this (eg, the Mini-Mental State Examination (MMSE) and Alzheimer's Disease Assessment Scale–cognitive subscale (ADAS-cog) for cognition, and the Texas Functional Living Scale and Barthel scale for activities of daily living).

We will assess the feasibility to conduct both traditional pairwise and network meta-analysis. Pooled estimates are calculated as weighted averages of the treatment effects in included studies, where weights are assigned to each study based on its precision.[43] Network meta-analysis extends pairwise meta-analysis by incorporating both direct evidence about the relative effectiveness of interventions that have been compared with each other in a primary study and indirect evidence about the relative effectiveness of interventions that were never directly compared with each other, but are connected through a network of other interventions for which direct comparisons exist. In addition, network meta-analysis can be a

valuable tool in delineating the effects of individual components of complex interventions, as recently shown in a network meta-analysis of multi-component interventions for dementia caregiver depression.[44] The validity of treatment effect estimates obtained through network meta-analysis depends on the similarity of the various direct comparisons in the network with respect to relevant treatment effect modifiers (eg, severity of disease, age of participants, risk of bias due to study design and implementation) and consistency of treatment effects obtained from direct and indirect comparisons.[45]

It is anticipated that in our sample of included studies, there will be underlying differences between individual studies of the same intervention, in particular with respect to the details of the intervention and the setting. For example, a training intervention in one study is unlikely to be exactly the same intervention as in another study, even if a protocol is used to standardise the intervention, because, for example, the individual delivering the training changes. Such differences do not preclude pooling study results, but the heterogeneity in treatment effects resulting from underlying differences needs to be taken into account when assigning weights to studies for a pooled estimate. Our default will therefore be to use a random-effects model, which takes into account between-study variation and assumes that included studies come from different populations, with unique details of the intervention and conditions of the study. The true treatment effect can therefore vary from study to study.[46]

## Subgroup analysis

While a random-effects model takes into account between-study variation, it does not explain heterogeneity. We will aim to use subgroup analysis and meta-regression to identify any study-level characteristics that might explain variation in treatment effects. Potential candidate covariates for subgroup analysis and meta-regression are study design and quality (within RCTs: risk of bias; within non-RCTs: analytical method employed, for example, instrumental variable, risk of bias), intervention details (eg, drug dose, intensity of training, intensity of stimulation therapy) and setting (by country, by bracket of gross national income, by rural vs urban setting). However, subgroup analysis will only produce meaningful insights when enough studies are included in each subgroup to detect any difference by the selected covariate (as opposed to a chance finding) and meta-regression should not be conducted for samples of less than 10 studies.[47] Given that we do not expect to identify a large number of robust dementia intervention studies in LMICs, it is unlikely that there will be sufficient studies of a given intervention with a specific outcome to allow meaningful subgroup analysis and meta-regression.

In cases where interventions were evaluated both by RCTs and non-randomised studies, we will obtain separate pooled effects for the two study types.[18] We will extract information on study design characteristics that will allow us to examine these as potential sources of heterogeneity.[38]

## Exploring meta-bias

For each intervention-outcome pair, we will first assess in a funnel plot whether asymmetry exists with respect to expected random variation of treatment effects around the pooled effect estimate with decreasing study precision. Funnel plots are used to detect possibly 'missing' smaller studies with larger or smaller treatment effects than what would be expected by chance alone, for which publication bias is one possible explanation.[48] We will then use contour-enhanced funnel plots to assess whether any observed asymmetry is likely due to publication bias favouring the publication of statistically significant results.[49]

We anticipate identifying only a small number of studies eligible for inclusion in our quantitative review (meeting the criteria of being conducted in a LMIC and using a robust study design). In a scenario where not more than one LMIC study exists for any combination of intervention and outcome, pairwise meta-analysis will become impossible. For network meta-analysis, the existence of only a handful of LMIC studies, even if a common comparator (eg, standard care) exists, makes it difficult to assess whether the assumptions of similarity (with respect to treatment effect modifiers) and consistency (agreement between direct and indirect estimates of treatment effect) hold. In case of a lack of eligible studies meeting our primary inclusion criteria, we will use our secondary set of inclusion criteria for studies using less robust research designs and will explore the feasibility of quantitative synthesis of these.

Given the reliance of past efforts to develop recommendations for dementia interventions in LMICs on evidence from high-income countries,[7 15] our primary aim is to advance our understanding of what works in LMICs and provide policymakers, people living with dementia and their carers, health and care professionals and others with evidence that is of immediate relevance to their setting. However, should quantitative synthesis of LMIC studies not be feasible due to a lack of studies reporting the same outcome, we will pursue an alternative way of obtaining pooled estimates of the comparative effectiveness of dementia interventions relevant for LMICs, as outlined below. In a scenario where we are unable to identify enough primary studies of dementia interventions in LMICs, we will quantitatively synthesise available evidence from methodologically robust studies from high-income countries, as identified in the MODEM Toolkit (www.modem-dementia.org.uk). We will discuss existing interventions with dementia experts from the STRiDE partner countries to identify those that have the highest relevance for LMICs. We will then extract relevant information from studies meeting our study design inclusion criteria in the MODEM database and obtain pooled estimates for the effectiveness and comparative effectiveness of these interventions. We will develop a separate protocol for this approach should it become necessary (ie, in the case of not identifying eligible studies in LMICs).

## Patient and public involvement

While no people living with dementia or carers were involved in the development of this protocol, the research project under which this systematic review will be undertaken (STRiDE), was developed in close collaboration with dementia advocacy groups and experts by experience. People living with dementia and carers will be involved in later stages of the systematic review process (selection of relevant interventions for wider dissemination as described below).

## ETHICS AND DISSEMINATION

No primary data collection will be conducted. We will include published reports of studies and will synthesise results of these at the aggregate level. We therefore did not seek ethics approval.

We plan to publish the findings of this systematic review both as a peer-reviewed article and in formats more accessible to people living with dementia, their carers, policymakers, care professionals and lay audiences, including on a website and DVD and in brief evidence summaries.

**Author affiliations**
[1]Personal Social Services Research Unit (PSSRU), London School of Economics and Political Science, London, UK
[2]Department of Health Policy, London School of Economics and Political Science, London, UK
[3]NIMHANS, Bangalore, India
[4]Federal University of Sao Paulo (UNIFESP), Sao Paulo, Brazil
[5]University of Nottingham Institute of Mental Health, Nottingham, UK
[6]Library, London School of Economics and Political Science, London, UK
[7]Personal Social Services Research Unit (PSSRU), University of Kent, Canterbury, UK
[8]Africa Mental Health Foundation, Nairobi, Kenya
[9]University of Nairobi, Nairobi, Kenya
[10]National Institute of Geriatrics, National Institutes of Health, Mexico City, Mexico

**Contributors** AC-H, DMD, HN and MS-K defined the scope of the review and review question. DMD, HN and MS-K developed inclusion/exclusion criteria. AF consulted on the databases to be searched and search terms used. MS-K drafted the protocol. MS-K, HN, DMD, SA, DO, AF, SH, MK, CM, DN, ML-O and AC-H reviewed the draft protocol along with the search strategy and contributed to a revised version. MS-K acts as the guarantor of the manuscript.

**Funding** This work is conducted as part of the 'Strengthening responses to dementia in developing countries' (STRiDE) project, supported by the UK Research and Innovation's Global Challenges Research Fund (ES/P010938/1). The funder was not involved in the development of this protocol at any stage.

**Competing interests** None declared.

**Patient consent for publication** Not required.

**Provenance and peer review** Not commissioned; externally peer reviewed.

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
