## [Reviewer comments · BMJ Open]

ARTICLE DETAILS

TITLE (PROVISIONAL)	Effectiveness of interventions for dementia in low- and middle-income countries: Protocol for a systematic review, pairwise and network meta-analysis
AUTHORS	Salcher-Konrad, Maximilian; Naci, Huseyin; McDaid, David; Alladi, Suvarna; Oliveira, Deborah; Fry, Andra; Hussein, Shereen; Knapp, Martin; MUSYIMI, CHRISTINE; Ndetei, David; Lopez-Ortega, Mariana; Comas-Herrera, Adelina

VERSION 1 - REVIEW

REVIEWER	Dr. Sandra Schüssler Medical University of Graz, Institute of Nursing Science, Austria
REVIEW RETURNED	30-Nov-2018

GENERAL COMMENTS	Dear Authors, following are my suggestions for the manuscript. In general, it is a very well written research protocol and the topic is of high international relevance. The research question “Which dementia interventions have been shown to be effective in low- and middle-income countries and how do they compare to each other?” including people with dementia, MCI and their caregivers (formal and informal) and care system/society/communities (financing of care; policy interventions, ...) is very broad for a systematic review. Am I right in assuming you include all settings (e.g. home care, community, day care centers, nursing homes, hospitals)? I would suggest to start first with a scoping review, because it is a very suitable type of review focusing on very broad research questions (like yours) to provide a good overview from the existing evidence. Based on the knowledge of the scoping review I would suggest you to start with a systematic review as a second step. Because then you can narrow your research question and a metaanalysis would be rather possible. I think with your very broad research question it is possible to get a very good overview, but at the same time it is not so easy to go into sufficient detail for every population and care setting. This may be a bias. However, if you have realized that there is only a very small amount of research available about different interventions in low- and middle-income countries, as you described, then it is justified to do it in the way you have described in your manuscript. Then a broader research question for a systematic review is alright.
---

	In Line 5, page 8: you state that you will focus on people already living with dementia and their carers and you will therefore exclude primary prevention studies. But in your population you will include people with MCI (Line 19/20, page 7)! Please describe this in more detail to make it clearer for your readers what you mean. Because your focus, beside other interventions, is also on technical interventions for dementia – I would suggest to include the IEEE database and, for grey literature and expert contact, to include social media databases.
--	---

REVIEWER	Rebecca Palm Deutsches Zentrum für neurodegenerative Erkrankungen (DZNE), Witten, Germany
REVIEW RETURNED	08-Feb-2019

GENERAL COMMENTS	Dear authors, I read your study protocol with great interest and really appreciate the in-depth reporting of planned methods and the clear style of writing. I have just one minor suggestion: please include why you plan to include non-experimental trials in the abstract. The explanation you give in the full-text is absolutely comprehensible, but the reader may question this when reading the abstract.
--

REVIEWER	Neil Chadborn University of Nottingham, UK
REVIEW RETURNED	15-Feb-2019

GENERAL COMMENTS	This is a brave approach to a challenging issue; effectiveness of interventions for care of people with dementia and their carers in Lower & Middle Income Countries. Reporting of interventions developed in LMIC is likely to be of poor quality due to resource issues and language difficulties. The various stakeholders and expertise with multiple languages indicates that this study has been well designed to meet the stated objectives. It is reassuring to read the complete PRISMA-P checklist, a summary of potential outcomes, and a full table of search terms and study selection criteria. The authors focus on experimental studies of any intervention with any type of dementia. The risk is that this may yield a heterogeneity of studies that is not appropriate for meta-analysis, and the authors have discussed this challenge and given contingency plans. Alternatively, the problem could have been approached as an implementation problem of cross-cultural adaptation of interventions which evidence of effectiveness in high income countries. Implementation studies are unlikely to be experimental (RCT) and therefore would be excluded from the proposed systematic review. It may be worth considering this approach early within the contingency plan; and consider updating the protocol at that stage. It seems disappointing that people with dementia or carers have not been involved in design of the study to date, nor are there any plans to do so. Considering the broad stakeholder network of the programme, it would seem appropriate to engage with members of the public as the review continues. This may be particularly
--

	important in terms of understanding the priority of various outcomes reported and also in making decisions about contingency plans. It is unclear why 2008 was chosen for a start date (presumably to give a 10 year period); could this be further justified? The language issue is addressed, especially with a diversity of languages spoken by the authors. However, I'm concerned that Chinese is not included and there may be a body of literature published in Chinese language. The authors should consider some scoping work to investigate whether there is a risk of missing literature from East Asia.
--	--

REVIEWER	Colleen Maxwell University of Waterloo, Canada
REVIEW RETURNED	15-Feb-2019

GENERAL COMMENTS	Generally a well written and described protocol. The authors address an extremely important area (dementia and care in LMICs) in need of further investigation. They have made a clear argument for the relevance of their protocol. The co-authors are involved with several other initiatives related to dementia care in LMICs and thus, have relevant experience and knowledge of the issues and resources specific to their research question. There are a few areas in need of further elaboration and/or consideration.  1. The main concern relates to the lack of specific focus (PICOS) in the research question - which is always an important consideration when embarking on a systematic review (with or without a meta-analysis). I do understand why the authors have allowed for such a general question for their protocol (given the concern about possibly finding no relevant studies) - however, it is challenging to understand how they will effectively summarize the existing literature given they are including ALL populations (those with dementia [any type] or MCI; carers [unpaid and paid - all types]), interventions, comparisons, outcomes, settings, etc. Further, just how effectiveness will be defined, and the nature of specific comparisons to be made - remain unclear given the breadth of potential studies (and study designs) to be included. There are comments throughout the protocol that suggest the authors are aware of the challenges they will face - however, further clarity is needed regarding how the authors will effectively summarize the diverse literature - including the implications that may span the individual, family and health care providers, health system, policy etc. 2. The authors need to clarify whether they will include studies of persons with MCI. In one part of the protocol they state they will - but later they state that they will only focus on those living with dementia (pg. 8). My suggestion would be to exclude studies of MCI - for obvious reasons (see point 1). 3. The authors wish to focus on studies with robust methods - and throughout the protocol they make the statement that the focus will be on studies that are capable of 'establishing causality'. There is some concern with this statement - as there are robust study designs / methods that strongly support causal inferences (but should never be interpreted as establishing causality). Some
---

	further clarity and specifics on the study designs to be included (and why) would be helpful. Also - some caution in using the phrase, 'establishing causality' is warranted. 4. The statement on pg. 7, "...be mindful of risks of dementia associated with conditions such as HIV/AIDS that may be more prevalent in LMICs." is unclear and requires further elaboration. What is meant by this statement? 5. On pg. 27, the authors state, "In a scenario where we are unable to identify enough primary studies of dementia interventions in LMICs, we will quantitatively synthesise available evidence from methodologically robust studies from high-income countries, as identified in the MODEM toolkit." This begs the question - why are the authors not doing this first? Some comment in this regard is needed - to help justify the jump to their current protocol.
--	--

VERSION 1 – AUTHOR RESPONSE

Reviewer 1:

In general, it is a very well written research protocol and the topic is of high international relevance.

We thank the reviewer for the positive assessment of our protocol and for the valuable comments provided.

The research question “Which dementia interventions have been shown to be effective in low- and middle-income countries and how do they compare to each other?” including people with dementia, MCI and their caregivers (formal and informal) and care system/society/communities (financing of care; policy interventions, ...) is very broad for a systematic review. Am I right in assuming you include all settings (e.g. home care, community, day care centers, nursing homes, hospitals)?

Indeed, we are not restricting this review to specific settings. However, we will record information about the setting of the study at the data extraction stage, as this will be important information when contextualising the results of included studies and for translating the findings of our systematic review into recommendations about “what works”.

I would suggest to start first with a scoping review, because it is a very suitable type of review focusing on very broad research questions (like yours) to provide a good overview from the existing evidence. Based on the knowledge of the scoping review I would suggest you to start with a systematic review as a second step. Because then you can narrow your research question and a metaanalysis would be rather possible. I think with your very broad research question it is possible to get a very good overview, but at the same time it is not so easy to go into sufficient detail for every population and care setting. This may be a bias. However, if you have realized that there is only a very small amount of research available about different interventions in low- and middle-income countries, as you described, then it is justified to do it in the way you have described in your manuscript. Then a broader research question for a systematic review is alright.

We thank the reviewer for this suggestion. Indeed, our proposed review has a broad scope. The aim of identifying and synthesising all available evidence on dementia interventions in LMICs (as long as it comes from robustly designed studies) was explicitly set in the context of this large, international research project. Ultimately, the project aims to provide a wide range of stakeholders (policy-makers, health care professionals, people living with dementia and their carers) with evidence summaries of what works to improve the lives of people living with dementia and their carers. We had initially

considered restricting the inclusion criteria to specific interventions, but realised that this approach would likely result in a biased selection of interventions that is informed by Western care pathways and standards of care. Other interventions might be more promising in less well-resourced settings, and we therefore deliberately opted not to restrict the review to specific interventions.

The reviewer also suggests conducting a scoping review before embarking on a systematic review of the literature. While we did not conduct a dedicated scoping review before developing the protocol for the systematic review, our approach was informed by previous research by our team and others (see below, as well as in the description of the rationale for including non-randomised studies on p8, para 2), as well as the consideration of the research consortium, including senior dementia researchers from seven low- and middle-income countries, that more robust evidence on the effectiveness of a wide range of dementia interventions in their countries is desired. The rationale for our approach was therefore as described by the reviewer: we found that relatively few studies in this setting were available and determined that a systematic review was important and feasible.

In a previous project ("MODEM"), members of this review team searched for, reviewed, and made available in an online database, evidence on a wide range of dementia interventions (available on <https://toolkit.modem-dementia.org.uk>). While that project's focus was on a high-income setting, the search strategy used to identify effective and cost-effective dementia interventions did not use geographical limiters and therefore also picked up studies from LMICs. Reviewing the studies included in the MODEM database, we found very few that were conducted in LMICs, therefore warranting a dedicated systematic review focusing on LMICs only (including LMIC-specific search terms and relevant databases). In addition, different groups often relied on studies conducted in high-income settings when developing packages of care for dementia in LMICs (Prince, MJ, et al. PLoS Med 2009;6:e1000176; Prince MJ, et al. World Alzheimer report 2016.), again indicating the value of conducting a systematic review of dementia interventions in this setting.

In our revised manuscript, we have referred to this in the introduction (p5, para 2) and pointed towards previous research and the need for a systematic review (changes underlined):

Interventions with proven positive impacts in high-income countries have previously been summarised in the MODEM dementia evidence toolkit. However, it is unclear whether interventions that showed beneficial effects in high-income settings, such as cognitive stimulation therapy for cognition and quality of life,[10,11] advance care planning,[12] training for formal carers (such as STAR)[13] and support for family carers (START),[14] are also effective in less-resourced settings, where awareness about dementia is lacking and dementia care may not be a policy priority. In this context, dementia is reported to be under-diagnosed and specialised care is often not available,[7] raising the question of which dementia interventions are effective in LMICs. Indeed, previously developed recommendations for packages of dementia care in LMICs were largely based on evidence from high-income countries,[Prince, MJ, et al. PLoS Med 2009;6:e1000176; Prince MJ, et al. World Alzheimer report 2016.] indicating a need to better understand what works in LMICs. The aim of this systematic review is to help fill this research gap by identifying dementia interventions for which robust evidence on effectiveness in LMICs exists and to synthesise available evidence to determine which interventions have the most potential of achieving desired outcomes in these settings.

In Line 5, page 8: you state that you will focus on people already living with dementia and their carers and you will therefore exclude primary prevention studies. But in your population you will include people with MCI (Line 19/20, page 7)! Please describe this in more detail to make it clearer for your readers what you mean.

The inconsistency was also picked up by reviewer 4 and our response therefore addresses concerns by both reviewers.

We will include both people living with dementia and those with MCI, due to the direct and strong link to developing dementia, as described on p6, para 2. We have now clarified that both groups are included in the sentence mentioned by the reviewer and have extended it to read as follows (p7, para 1; changes underlined):

We will focus on people already living with dementia or MCI and their carers and will therefore exclude primary prevention studies (i.e. prevention of dementia in people without cognitive impairment).

Because your focus, beside other interventions, is also on technical interventions for dementia – I would suggest to include the IEEE database and, for grey literature and expert contact, to include social media databases.

We thank the reviewer for these constructive suggestions. We had previously considered searching additional grey literature data bases to the ones already included, but concluded that our search strategy, encompassing 9 academic and grey literature databases, as well as expert consultation and citation searching, was comprehensive enough to capture relevant studies. We decided against searching additional data bases due to an expected low yield of studies meeting our inclusion criteria. Nevertheless, we ran an exploratory search of the IEEE data base but found this to contain mostly proof-of-concept studies in dementia, which, while providing an interesting outlook into the future, do not meet our criteria of robust study designs.

Reviewer 2:

Dear authors, I read your study protocol with great interest and really appreciate the in-depth reporting of planned methods and the clear style of writing. I have just one minor suggestion: please include why you plan to include non-experimental trials in the abstract. The explanation you give in the full-text is absolutely comprehensible, but the reader may question this when reading the abstract.

We thank the reviewer for the positive feedback. To address the minor point raised, we have included a sentence describing the rationale for including both randomised and non-randomised studies in the abstract (p2):

We will include both randomised and non-randomised studies due to an anticipated low number of well-conducted randomised trials in LMICs and potentially greater external validity of non-randomised studies conducted in routine care settings.

Reviewer 3:

This is a brave approach to a challenging issue; effectiveness of interventions for care of people with dementia and their carers in Lower & Middle Income Countries. Reporting of interventions developed in LMIC is likely to be of poor quality due to resource issues and language difficulties. The various stakeholders and expertise with multiple languages indicates that this study has been well designed to meet the stated objectives. It is reassuring to read the complete PRISMA-P checklist, a summary of potential outcomes, and a full table of search terms and study selection criteria.

We thank the reviewer for the feedback provided and are pleased to see that specific strengths of our review have been noted, namely the inclusion of various stakeholders, reviewers fluent in a range of languages, and the thorough design of the systematic review protocol in compliance with good practice for this type of research.

The authors focus on experimental studies of any intervention with any type of dementia. The risk is that this may yield a heterogeneity of studies that is not appropriate for meta-analysis, and the authors have discussed this challenge and given contingency plans. Alternatively, the problem could have been approached as an implementation problem of cross-cultural adaptation of interventions which evidence of effectiveness in high income countries. Implementation studies are unlikely to be experimental (RCT) and therefore would be excluded from the proposed systematic review. It may be worth considering this approach early within the contingency plan; and consider updating the protocol at that stage.

We thank the reviewer for raising a question about the potentially limited relevance of experimental studies (RCTs) for the implementation of dementia interventions. We agree with the reviewer that experimental studies, due to their relatively narrow inclusion criteria and experimental set-up, are an essential, but not the only piece of evidence when aiming to understand what interventions work. It is for this reason, as well as the anticipated lack of well-conducted RCTs in the LMIC setting, that our inclusion criteria specify eligible study designs as “experimental designs (RCTs and cluster-RCTs) as well as non-randomised designs suitable for causal inference” (p8, para 2). We have laid out our rationale for including both randomised and non-randomised studies in detail on pages 8-10 of the manuscript. This section of the manuscript also states that we are planning to include study designs that are particularly suited for evaluation of interventions in a non-experimental setting (specifically, controlled before-after studies, interrupted time series studies, and difference-in-differences studies; p9, para 1).

We also agree with the reviewer that a contingency plan was necessary in case our primary study design eligibility criteria are not met by any study, which was already provided on p9, para 2 of the manuscript.

It seems disappointing that people with dementia or carers have not been involved in design of the study to date, nor are there any plans to do so. Considering the broad stakeholder network of the programme, it would seem appropriate to engage with members of the public as the review continues. This may be particularly important in terms of understanding the priority of various outcomes reported and also in making decisions about contingency plans.

We fully agree with the reviewer about the importance of engaging with people living with dementia and carers. The statement included in the previous version of the manuscript about lack of involvement was only valid for the development of the systematic review protocol. We have now revised this paragraph to describe how people living with dementia were involved in the development of the wider project (STRiDE) and their planned involvement at later stages of the review (p26, para 2, changes underlined):

While no people living with dementia or carers were directly involved in the development of this protocol, the research project under which this systematic review will be undertaken (STRiDE), was developed in close collaboration with dementia advocacy groups and experts by experience. People living with dementia and carers will be involved in later stages of the systematic review process (selection of relevant interventions for wider dissemination as described below).

It is unclear why 2008 was chosen for a start date (presumably to give a 10 year period); could this be further justified?

Indeed, the year 2008 was initially chosen to cover a 10-year period when we first started working on the protocol (Spring 2018). The 10-year period was chosen to identify interventions that would still be relevant today. By the time we submitted the protocol, we had updated this to the end of 2018, therefore the period is now covering a total of 11 years.

The language issue is addressed, especially with a diversity of languages spoken by the authors. However, I'm concerned that Chinese is not included and there may be a body of literature published in Chinese language. The authors should consider some scoping work to investigate whether there is a risk of missing literature from East Asia.

The reviewer raises an important point which we had considered during the preparatory phase for this protocol. Recognising the vast amount of Chinese literature available, we attempted to translate our search strategy and ran pilot searches of a key Chinese data base (CNKI – Chinese National Knowledge Infrastructure). However, we found the inclusion of Chinese data bases not to be feasible within the resource restraints of our project due to a) the complexities in translating the existing search terms into Chinese, even with the help of native speakers, and b) the vast amount of records (~80,000) that even simple searches of CNKI retrieved. We consider a systematic review of the Chinese literature on dementia interventions a worthwhile undertaking on its own right but are unable to include this vast exercise in our review. However, recognising the importance of the Chinese literature, we are pleased to report that outreach from our project to additional countries has led to a partner institution in Hong Kong taking up the task of reviewing the Chinese literature on dementia interventions. We will be collaborating on this separate systematic review of Chinese language studies, which we hope will complement our findings.

It is also important to note that several strategies will ensure that key papers in Chinese will be captured in our review. First, we recognise the selection of databases in selected languages as a potential limitation. However, our search strategy is comprehensive and captures studies in various languages, including Chinese. Our search strategy includes not only database searches, but also capitalises on the expertise of researchers in LMICs (research consortium partners and extended network, including the above-mentioned collaboration with a research institute in Hong Kong), and we are therefore confident that important studies will be retrieved even if not indexed in databases.

Second, we had previously only listed the languages spoken by members of the core review team in the protocol, but we will capitalise on the research project's extensive international network to also review studies in other languages, including studies from China. We have now included Chinese as one of the languages spoken by the review team, as we are able to review studies from China through the above mentioned partnership.

We have included a sentence on the expanded network of our consortium in the revised manuscript (p13, para 1) and have included Chinese among the languages spoken by the review team (Box 1, p11, bullet 5; changes underlined):

We will also capitalise on the expertise of STRiDE consortium partners (researchers and dementia advocacy groups) in seven LMICs (Brazil, India, Indonesia, Jamaica, Kenya, Mexico, South Africa), and an expanded network of STRiDE collaborators in other countries, to include published and unpublished studies from these and other countries meeting our eligibility criteria if they were not captured by database searches.

Box 1 (exclusion criteria):

- Studies where reports are not available in a language spoken by the international project team members (languages spoken include Arabic, Chinese, English, French, German, Bahasa Indonesia, Portuguese, Romanian, Spanish, Turkish, and others)

Reviewer 4:

Generally a well written and described protocol.

The authors address an extremely important area (dementia and care in LMICs) in need of further investigation. They have made a clear argument for the relevance of their protocol. The co-authors are involved with several other initiatives related to dementia care in LMICs and thus, have relevant experience and knowledge of the issues and resources specific to their research question.

There are a few areas in need of further elaboration and/or consideration.

We thank the reviewer for the overall positive assessment of our protocol and for taking time to consider in detail areas that can be strengthened. Point-by-point responses to the issues raised are provided below.

1. The main concern relates to the lack of specific focus (PICOS) in the research question - which is always an important consideration when embarking on a systematic review (with or without a meta-analysis). I do understand why the authors have allowed for such a general question for their protocol (given the concern about possibly finding no relevant studies) - however, it is challenging to understand how they will effectively summarize the existing literature given they are including ALL populations (those with dementia [any type] or MCI; carers [unpaid and paid - all types]), interventions, comparisons, outcomes, settings, etc.

Further, just how effectiveness will be defined, and the nature of specific comparisons to be made - remain unclear given the breadth of potential studies (and study designs) to be included.

There are comments throughout the protocol that suggest the authors are aware of the challenges they will face - however, further clarity is needed regarding how the authors will effectively summarize the diverse literature - including the implications that may span the individual, family and health care providers, health system, policy etc.

We thank the reviewer for raising these important points in relation to the evidence synthesis stage of the review. Indeed, the protocol does not specify in detail which comparisons will be made (which we have now added, as described below). As acknowledged by the reviewer, there is a rationale behind keeping the inclusion criteria relatively broad. Since it is the aim of the review to synthesise all available evidence (coming from robust studies) on the effectiveness of dementia interventions, we are unable to predict which specific comparisons will be made (this will depend on the interventions identified through the review process and the outcomes measured in included studies), or define an overall valid definition for effectiveness (this will depend on the outcomes measured in included studies). Nevertheless, we have given thought to the synthesis process and describe in the protocol how we are planning to analyse included studies and summarise the diverse literature. Specifically, on p23, para 2, we state how the evidence will be summarised along several dimensions, including the "target" group of the intervention, aim of the intervention, intervention type, and outcomes, and this section has now been expanded (see below). We have also presented a potential categorisation of studies according to outcomes and stakeholder groups in the online supplement, and p23 para 3 describes that this is a draft framework subject to change after completion of data extraction.

We have now expanded p23, para 2 to clarify the narrative synthesis part of the review and how evidence is presented in more granular form for informed decision-making. P23 now reads as follows (changes underlined):

We will first describe the interventions that have been evaluated in rigorous study designs in LMICs and summarise the features of the intervention, where they were studied, characteristics of the studies, and their findings. We are planning to tabulate interventions according to who they are targeted at (person living with dementia, their carers, care professionals and care systems), aim of the intervention (secondary prevention, treatment, disease management, coping with the disease and caring for people living with dementia, managing the impact of dementia on the care system), intervention type (including, but not limited to, pharmaceuticals, cognitive therapy, technological

interventions, exercise, training, diagnosis, organisation of care, financing of care, policy interventions), and outcomes studied. We plan to review details extracted from included studies on the population, intervention, and outcomes (see Box 3) to group studies accordingly. For example, in terms of population, included studies may be grouped according to severity of impairment (for people living with dementia) and type of carer (unpaid, paid, professional, family caregiver). Tabulation along this dimension will allow us to explore which types of interventions were shown to be effective for each of the groups in order to provide sufficiently granular findings to inform decision-making in specific contexts. Similarly, we plan to tabulate studies according to the aim of the intervention, intervention type, and outcome studied, thereby providing a summary of the evidence for different aspects of dementia interventions.

Potentially relevant outcomes for this systematic review can be characterised by their type and the stakeholder group they refer to. A draft list of potentially relevant outcomes and their stakeholder groups according to which our analysis can be structured is provided in the online supplementary table. The list is subject to change after reviewing the outcomes used in included studies, which might use other outcomes that could necessitate a different categorisation.

2. The authors need to clarify whether they will include studies of persons with MCI. In one part of the protocol they state they will - but later they state that they will only focus on those living with dementia (pg. 8). My suggestion would be to exclude studies of MCI - for obvious reasons (see point 1).

We have now resolved this inconsistency (see response to reviewer 1).

We will include both people living with dementia and those with MCI, due to the direct and strong link to developing dementia, as described on p6, para 2. We have now clarified that both groups are included in the sentence mentioned by the reviewer and have extended it to read as follows (p7, para 1; changes underlined):

We will focus on people already living with dementia or MCI and their carers and will therefore exclude primary prevention studies (i.e. prevention of dementia in people without cognitive impairment).

3. The authors wish to focus on studies with robust methods - and throughout the protocol they make the statement that the focus will be on studies that are capable of 'establishing causality'. There is some concern with this statement - as there are robust study designs / methods that strongly support causal inferences (but should never be interpreted as establishing causality). Some further clarity and specifics on the study designs to be included (and why) would be helpful. Also - some caution in using the phrase, 'establishing causality' is warranted.

We agree with the reviewer that appropriate language is warranted when describing the causal inference of treatment effects obtained from non-randomised studies. We have now revised the manuscript to use more conservative language in relation to the causal attribution of treatment effects in these studies throughout the protocol (see below).

We have given considerable thought to the types of study designs eligible for inclusion and have described the eligibility criteria on p8-10 of the manuscript, focusing in particular on non-randomised studies. As described on p8, para 3, the eligibility of non-randomised studies will not be guided by specific study labels, which are used inconsistently and are therefore not recommended as eligibility criteria. Instead, we will use a series of questions, proposed by Cochrane Non-Randomized Studies for Interventions Methods Group, and reproduced in Table 1.

We have laid out the reasons for also including non-randomised studies on p8, para 2, which include an anticipated low number of RCTs in LMICs, as well as wanting to capture evidence from non-experimental settings. The study designs we have mentioned as examples (on p9, para 1) include those particularly suited for evaluation in routine care settings (specifically, controlled before-after

studies, interrupted time series studies, and difference-in-differences studies; see also our response to the second point made by reviewer 3 above).

We made several changes throughout our revised manuscript to substantially tone down the 'causality' terminology to address the reviewer's comment. Our changes are highlighted in the following sections (changes underlined):

Abstract:

Studies will be eligible for inclusion if they were published between 2008 and 2018, conducted in LMICs and evaluated the effectiveness of a dementia intervention using a study design that supports causal inference of the treatment effect.

Strengths and limitations of the study (p3, bullet 2):

Eligible study designs include both randomised trials and non-randomised studies supporting causal inference of treatment effects, with stringent eligibility criteria applied for the latter.

P5, para 1:

Evidence-based practices are informed by studies with strong research designs supporting a causal link between an intervention (be it a drug, non-pharmacological therapy, organisational change, or another form of intervention) and improved outcomes in people living with dementia and their carers. While randomised controlled trials (RCTs) are considered the 'gold standard' of intervention studies, other study designs exist that allow researchers to draw causal conclusions about the effect of an intervention in the absence of RCTs and can provide essential evidence on 'what works'.^[8,9]

P8, para 2:

The research question we aim to answer is what works in dementia in LMICs. We are therefore interested in the effect that an intervention had on outcomes in people receiving the intervention, their carers and the wider care system.^[19] Accordingly, we will apply a causal inference framework in this review: study designs eligible for inclusion are those that support a causal link between the intervention and observed outcomes. This includes experimental designs (RCTs and cluster-RCTs) as well as non-randomised designs suitable for causal inference, defined as "comparisons of 'potential outcomes' [...] under different treatment conditions on a common set of units."^[8]

P8, para 3:

The label 'quasi-experimental' is discipline-specific and members of the Cochrane Non-Randomized Studies for Interventions Methods Group therefore developed a 'label-free' taxonomy of quasi-experimental studies that aims to define such studies through a series of questions in six domains.^[22] We will use their checklist to guide the selection of studies by study design (see Table 1). Since we anticipate a shortage of methodologically robust studies of dementia interventions in LMICs, but still aim to synthesise the best available evidence that supports causal inference of treatment effects, we adapted the original checklist by removing restrictions to answer 'yes' to only one question in domains 2, 3, and 4.

P14, para 3:

For identification of non-randomised studies that support causal inference of the relationship between exposure and outcome, we created an additional set of search terms pertaining to commonly used labels for quasi-experimental studies.

Box 3 on p22, bullet 4:

Study design: description and coding of the study design and how causality of a treatment effect was supported, including which covariates were used in the analysis to control for confounding and judgement about whether criteria for causality for the specific study design were met (in the case of non-randomised studies).

4. The statement on pg. 7, "...be mindful of risks of dementia associated with conditions such as HIV/AIDS that may be more prevalent in LMICs." is unclear and requires further elaboration. What is meant by this statement?

We have revised the manuscript to clarify that we took these risks into consideration while developing the search strategy. The relevant paragraph (p6, para 2) now reads as follows (changes underlined):

We will include studies in all stages (including early- to late-stage dementia) and different forms of dementia (e.g., Alzheimer's disease, vascular dementia, Lewy bodies dementia, and mixed dementias, as well as less common forms such as fronto-temporal dementia). We will also take into consideration risks of dementia associated with conditions such as HIV/AIDS that may be more prevalent in LMICs to ensure these are captured in our search strategy.

5. On pg. 27, the authors state, "In a scenario where we are unable to identify enough primary studies of dementia interventions in LMICs, we will quantitatively synthesise available evidence from methodologically robust studies from high-income countries, as identified in the MODEM toolkit." This begs the question - why are the authors not doing this first? Some comment in this regard is needed - to help justify the jump to their current protocol.

As laid out in the introduction section of the manuscript, the LMIC context appears very different from high-income countries: lower diagnosis rates of dementia, fewer resources available, overall lower awareness among the public and policy-makers. It is unclear whether a) interventions that work in high-income settings work in LMIC, and b) whether other interventions than those studied in high-income settings are available and proven to be effective. Our primary aim is to provide evidence that is of immediate relevance to the setting which we are focusing on (LMICs). Translating the findings from high-income settings to a LMIC setting comes with its own challenges (what factors to adjust for and how?) and our preference is to obtain robust evidence from the LMIC setting directly. Having reviewed a sample of appr. 500 abstracts of records identified in an initial database search, we are confident that we will identify a considerable number of relevant studies from LMICs.

Realising that our reasoning may not have been sufficiently clear in the old version of the manuscript, we have now added text to the relevant paragraph (p26, para 1, changes underlined):

Given the reliance of past efforts to develop recommendations for dementia interventions in LMICs on evidence from high-income countries,[Prince, MJ, et al. PLoS Med 2009;6:e1000176; Prince MJ, et al. World Alzheimer report 2016.] our primary aim is to advance our understanding of what works in LMICs and provide policy-makers, people living with dementia and their carers, health and care professionals and others with evidence that is of immediate relevance to their setting. However, should quantitative synthesis of LMIC studies not be feasible due to a lack of studies reporting the same outcome, we will pursue an alternative way of obtaining pooled estimates of the comparative effectiveness of dementia interventions relevant for LMICs, as outlined below.

VERSION 2 – REVIEW

REVIEWER	Dr. Sandra Schüssler Medical University of Graz, Austria
REVIEW RETURNED	15-Apr-2019

GENERAL COMMENTS	Dear authors, all revisions were performed satisfactorily. I have no further recommendations.
---

REVIEWER	Neil Chadborn University of Nottingham, UK
REVIEW RETURNED	12-Apr-2019

GENERAL COMMENTS	Thanks for responding to comments.
------------------------------------

REVIEWER	Colleen Maxwell University of Waterloo Canada
REVIEW RETURNED	02-May-2019

GENERAL COMMENTS	The authors have done a very thorough job in addressing all of the reviewers' comments. I appreciate their attention and thoughtfulness in all their responses. I have no further issues and wish the team all the best with this important work.
---